# Fractional Derivative Viscosity of ANCF Cable Element

**Yaqi Gu** [1] , **Zuqing Yu** [2], **Peng Lan** [3] **and Nianli Lu** [1,*]

1    School of Mechatronics Engineering, Harbin Institute of Technology, Harbin 150001, China
2    College of Mechanical and Electrical Engineering, Hohai University, Changzhou 213022, China
3    School of Mechanical & Electrical Engineering, Xi'an University of Architecture and Technology,
     Xi'an 710055, China
*    Correspondence: n.lu@hit.edu.cn

**Abstract:** Typical engineering cable structures, such as high-voltage wire and wire rope, usually bring a damping effect which cannot be ignored due to the technological problems of manufacturing. For such problems, especially the damping of cable structures undergoing large displacement and severe deformation, few studies have been reported in the past. In this work, the fractional derivative viscosity model is introduced into the cables described by the absolute nodal coordinate formulation. The computer implementation algorithm of the proposed cable damping model is given based on the three-parameter fractional derivative model. Two numerical examples demonstrate the effectiveness and convergence property of the proposed cable damping model. An experiment is proposed in which a wire is tensioned and released. Configurations are captured by the high-speed camera and compared with the results obtained from the numerical simulation. The agreement of the simulation and experimental results validates the proposed cable damping in application.

**Keywords:** fractional derivative viscosity; cable damping; absolute nodal coordinate formulation; flexible multibody system dynamics

## 1. Introduction

Cables are widely used in engineering applications such as electricity [1], cranes [2], cable-driven manipulators [3], deployable antennas [4], etc. In such systems, the dynamic simulation usually needs to face problems including large deformation and overall motion, material nonlinearity, impact and contact, etc. In particular, cables usually work with other rigid components to form a rigid-flexible coupled multibody system. Therefore, a sophisticated method is demanded to describe the motion of the cables.

The development of flexible multibody system dynamics has undergone the stages of the finite section method, the finite element method, the floating frame reference method, the geometrically exact method and the absolute nodal coordinate formulation (ANCF) method. Among these modeling tools, ANCF can be seen as a combination of continuous mechanics and the finite element method and is suitable for large deformation problems [5]. It has the advantages of a constant mass matrix and zero centrifugal and Coriolis forces. There has been a rich element library and increasing engineering applications based on ANCF. The spatial cable element, which is suitable for modeling cables, was first developed by Sugiyama et al. [6]. It uses the Euler–Bernoulli beam assumption, so the deformation of the cross-section and the shear deformation are omitted. Only the position and gradient vectors at two ends are preserved as nodal coordinates [7]. As fewer nodal coordinates and integration points are used, the cable element usually exhibits higher efficiency in computer implementation. Wang et al. discussed the contact between two or more ropes represented by the ANCF cable element. The beam-to-beam contact was performed by assuming a continuous contact zone and executing integration in the contact zone [8,9]. Li et al. utilized the ANCF cable element in the dynamic analysis of a deployable mesh reflector of a satellite antenna. A parallel computation algorithm was proposed and used in a deployment

simulation [10,11]. Lan et al. investigated the application of the ANCF cable element in the high-voltage wire modeling. The pre-tension algorithm and the non-linear sliding joints between the wire and the iron tower are studied [12]. Gu et al. studied the dynamic interaction between the high-voltage wire and the large-scale steel structure [1]. Boumann and Bruckmann proposed an emergency strategy for cable failure in cable robots [13]. Bulin and Hajzman developed an efficient approach for the non-linear elastic forces of the ANCF cable element by the pre-computation of various terms that are constant when evaluated numerically at Gaussian points [14]. Fotland and Haugen examined the performance of the Runge–Kutta method and the generalized alpha method in the dynamical simulation of the ANCF cable element [15]. It can be summarized that the ANCF cable element is capable of describing the complex deformation of the cable, which is the main reason for its successful application in engineering.

In engineering practice, both the steel rope and high-voltage transmission wire are manufactured by twisting metal threads. Due to the friction between the threads, they usually exhibit certain damping characteristics under intense dynamic behavior. Many efforts have been made to investigate the damping effect of the cable in civil engineering [16], deployable space structure [17], and so on. It is reported that the vibration characteristics of the cable can significantly affect the performance of the whole system [18]. As an advantage of ANCF, it is convenient to add material damping into the multibody system dynamic equations. In research reported by Garcia-Vallejo et al., the internal damping based on linear viscoelasticity in multibody system simulations performed via ANCF was investigated [19]. Lee et al. investigated the damping characteristics of a flexible multibody under different working conditions [20]. Kim reported their experimental research of two damping models based on ANCF [21]. In an article published in 2011, a non-linear viscoelastic constitutive model was introduced to the flexible multibody system represented by ANCF [22]. Grossi and Shabana researched high-frequency modes based on ANCF [23]. A new objective large rotation and large deformation viscoelastic constitutive model defined by the Navier–Stokes equations and its implicit numerical integration algorithm was proposed. Yu et al. studied the viscoelastic beam element via ANCF and used it in the dynamic analysis of two-link flexible manipulators [24]. Tian et al. proposed a new viscoelastic ANCF solid element to model the components of dielectric elastomers [25].

Essentially, the abovementioned material damping models are based on the Maxwell model and the Kelvin-Voigt constitutive model. The damping force is related to the changing rate of the strain, which makes these models cumbersome in computer implementation, as there are lots of coefficients that need to be calibrated [26]. It should be pointed out that there is a new kind of calculus in which the derivative order could be a fractional number. In recent years, a fractional-order derivative was gradually used to deal with numerous engineering problems, due to its indispensable role in the research of dynamic behavior, system optimization and other engineering problems. The applications of fractional-order derivatives in engineering problems are mainly in the fields of dynamics and control [27]. In the field of dynamics, the fractional-order derivative was commonly used to model engineering materials with memory properties, such as viscoelastic components. By defining more accurate constitutive relations of materials, the vibration characteristics of non-linear systems can be more reasonably analyzed [28]. With the development of numerical calculation methods, fractional derivative damping materials have been used in multibody system dynamics with numerical calculation as the main simulation method [29]. Zhang et al. introduced the fractional derivative damping model to the flexible multibody system dynamic simulation [30]. Lan et al. combined the fractional derivative viscosity with the ANCF thin plate element and applied it in tire modeling [31]. It was reported that compared with the traditional Maxwell model and the Kelvin–Voigt constitutive model, the damping model which assumed the viscous force related to the fractional derivative of the strain could reflect the dynamic behavior of the viscoelastic material more accurately [32].

From the summary above, one can find that the implementation of the fractional derivative material damping model in the ANCF cable element still remains blank, which

will be the main, original work of this investigation. The performance of this damping constitutive model will be checked both numerically and experimentally. The algorithm presented in this research will provide a more sophisticated benchmark in cable viscosity and can be used in engineering applications such as the dynamic analysis of the electric wire and steel rope-pulley system. The following part of this paper will be organized as follows: Sections 2 and 3 give a brief review of the kinematic description and elastic force formulation of the ANCF cable element, which are the bases of the proposed damping model. In Section 4, the implementation of the fractional-order derivative material damping in the ANCF cable element is given in detail. Section 5 discusses the computer implementation of the system equation of motion. Numerical and experimental results are given in Section 6, whereas Section 7 is the concluding remarks.

## 2. Element Kinematic Description

The ANCF cable element shown in Figure 1 is developed based on the Euler–Bernoulli beam model. The cross-section of the element is assumed to be undeformed and perpendicular to the centerline. Therefore, only the axial material coordinate of the centerline is used in the element shape function. The global position and gradient vectors of two end-points are taken as nodal coordinates so the total degree of freedom is 12. The global position of an arbitrary point on the element is defined as $\mathbf{r} = \mathbf{S}\mathbf{e}$. $\mathbf{e} = \left[\mathbf{r}^{i^T}, \mathbf{r}_x^{i^T}, \mathbf{r}^{j^T}, \mathbf{r}_x^{j^T}\right]^T$ is the element nodal coordinate vector, and the shape function is:

$$\mathbf{S} = [S_1\mathbf{I}_{3\times3}, S_2\mathbf{I}_{3\times3}, S_3\mathbf{I}_{3\times3}, S_4\mathbf{I}_{3\times3}] \tag{1}$$

where the components of the element shape function matrix are:

$$\begin{cases} S_1 = 1 - 3\xi^2 + 2\xi^3, & S_2 = l\left(\xi - 2\xi^2 + \xi^3\right) \\ S_3 = 3\xi^2 + 2\xi^3 & , & S_4 = l\left(-\xi^2 + \xi^3\right) \end{cases} \tag{2}$$

In this equation, the dimensionless material coordinate $\xi = x/l$. $l$ is the element length of its reference configuration. The element mass matrix has the following expression:

$$\mathbf{M} = \int_V \rho \mathbf{S}^T \mathbf{S}\,\mathrm{d}V \tag{3}$$

It can be found that the mass matrix of the flexible body described by ANCF is constant. The matrix decomposition can be performed at the pre-processing stage to improve simulation efficiency.

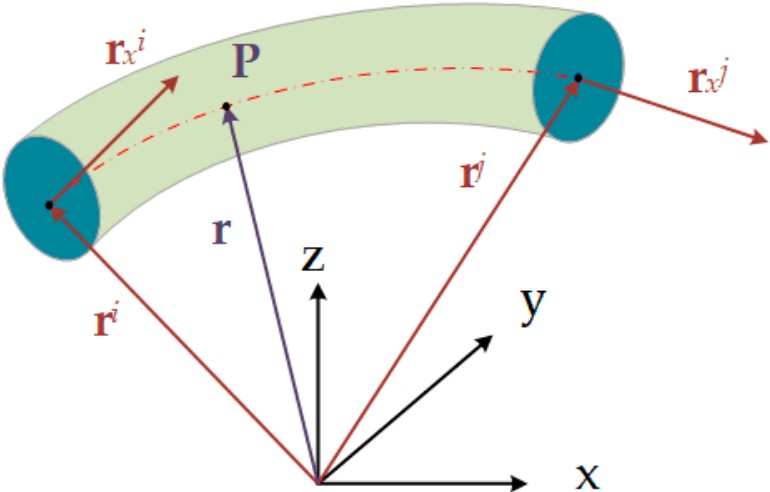

**Figure 1.** ANCF cable element.

## 3. Element Elastic Model

The strain energy of an ANCF cable element is constituted by the stretch and the curvature of the centerline [33]:

$$U = \frac{1}{2}\int_0^l EA(\varepsilon_{xx})^2 \mathrm{d}x + \frac{1}{2}\int_0^l EI(K)^2 \mathrm{d}x \tag{4}$$

In this equation, $l$ is the length of the element, $E$ is the modulus of elasticity, $A$ is the cross-section area and $I$ is the second moment of the cross-section. $\varepsilon_{xx}$ is the strain at the longitudinal direction, which has the following definition:

$$\varepsilon_{xx} = |\mathbf{r}_x| - 1 \tag{5}$$

$K$ is the spatial measurement of curvature:

$$K = \frac{|\mathbf{r}_x \times \mathbf{r}_{xx}|}{|\mathbf{r}_x|^2} \tag{6}$$

Then, the element generalized elastic force can be determined by deriving the strain energy with respect to the nodal coordinate:

$$\mathbf{Q}_e = \frac{\partial U}{\partial \mathbf{e}} = \int_0^l EA\varepsilon_{xx}\frac{\partial \varepsilon_{xx}}{\partial \mathbf{e}}dx + \int_0^l EIK\frac{\partial K}{\partial \mathbf{e}}dx \tag{7}$$

In this equation, the partial derivative of the longitudinal strain with respect to the $i$-th component of the element nodal coordinate vector is:

$$\frac{\partial \varepsilon_{xx}}{\partial e_i} = \frac{\partial(|\mathbf{r}_x| - 1)}{\partial e_i} = \frac{1}{|\mathbf{r}_x|}\mathbf{S}_{x,i}^{\mathrm{T}} \cdot \mathbf{r}_x \tag{8}$$

where $\mathbf{S}_{x,i}$ is the $i$-th column of the partial derivative of the element shape function with respect to the material coordinate. Denote $f = |\mathbf{r}_x \times \mathbf{r}_{xx}|$ and $g = |\mathbf{r}_x|^2$; the partial derivative of the curvature measurement is:

$$\frac{\partial K}{\partial e_i} = \frac{\partial}{\partial e_i}\left(\frac{f}{g}\right) = \frac{1}{g^2}\left(\frac{\partial f}{\partial e_i}g - f\frac{\partial g}{\partial e_i}\right) \tag{9}$$

In the equation above, the derivatives of the numerator f and denominator g are:

$$\begin{cases} \dfrac{\partial f}{\partial e_i} = \dfrac{1}{f}(\mathbf{S}_{x,i} \times \mathbf{r}_{xx} + \mathbf{r}_x \times \mathbf{S}_{xx,i})^{\mathrm{T}} \cdot (\mathbf{r}_x \times \mathbf{r}_{xx}) \\ \dfrac{\partial g}{\partial e_i} = 2 \cdot \mathbf{S}_{x,i}^{\mathrm{T}} \cdot \mathbf{r}_x \end{cases} \tag{10}$$

where $\mathbf{S}_{xx,i}$ is the $i$-th column of the second-order partial derivative of the element shape function. If the implicit integrator is used to solve the system equation of motion, the Jacobian matrix J of the elastic force with respect to nodal coordinates should be calculated:

$$\begin{aligned} J_{i,j} &= \frac{\partial^2 U}{\partial e_i \partial e_j} \\ &= \int_0^l EA\frac{\partial \varepsilon_{xx}}{\partial e_i}\frac{\partial \varepsilon_{xx}}{\partial e_j}dx + \int_0^l EA\varepsilon_{xx}\frac{\partial^2 \varepsilon_{xx}}{\partial e_i \partial e_j}dx + \int_0^l EI\frac{\partial K}{\partial e_i}\frac{\partial K}{\partial e_j}dx + \int_0^l EAK\frac{\partial^2 K}{\partial e_i \partial e_j}dx \end{aligned} \tag{11}$$

In this equation, the second derivative items can be formulated as:

$$\frac{\partial^2 \varepsilon_{xx}}{\partial e_i \partial e_j} = \frac{\partial}{\partial e_j}\left(\frac{1}{|\mathbf{r}_x|}\mathbf{S}_{x,i}^{\mathrm{T}} \cdot \mathbf{r}_x\right) = \frac{1}{|\mathbf{r}_x|}\left(\mathbf{S}_{x,i}^{\mathrm{T}} \cdot \mathbf{S}_{x,j} - \mathbf{S}_{x,j}^{\mathrm{T}} \cdot \mathbf{S}_{x,i}\right) \tag{12}$$

and:

$$
\begin{aligned}
\frac{\partial^2 K}{\partial e_i \partial e_j} &= \frac{\partial}{\partial e_j}\left(g^{-1}\frac{\partial f}{\partial e_i} - fg^{-2}\frac{\partial g}{\partial e_i}\right) \\
&= -\frac{1}{g^2}\frac{\partial g}{\partial e_j}\frac{\partial f}{\partial e_i} + \frac{1}{g}\frac{\partial^2 f}{\partial e_i \partial e_j} - \frac{1}{g^2}\frac{\partial f}{\partial e_j}\frac{\partial g}{\partial e_i} + \frac{2f}{g^3}\frac{\partial g}{\partial e_j}\frac{\partial f}{\partial e_i} - \frac{f}{g^2}\frac{\partial^2 g}{\partial e_i \partial e_j}
\end{aligned}
\tag{13}
$$

where the second derivatives of $f$ and $g$ are:

$$
\begin{aligned}
\frac{\partial^2 f}{\partial e_i \partial e_j} &= -\frac{1}{f^2}\frac{\partial f}{\partial e_j}\left(\mathbf{S}_{x,i} \times \mathbf{r}_{xx} + \mathbf{r}_x \times \mathbf{S}_{xx,i}\right)^{\mathrm{T}} \cdot \left(\mathbf{r}_x \times \mathbf{r}_{xx}\right) \\
&\quad + \frac{1}{f}\left(\mathbf{S}_{x,i} \times \mathbf{S}_{xx,j} + \mathbf{S}_{x,j} \times \mathbf{S}_{xx,i}\right)^{\mathrm{T}} \cdot \left(\mathbf{r}_x \times \mathbf{r}_{xx}\right) \\
&\quad + \frac{1}{f}\left(\mathbf{S}_{x,i} \times \mathbf{r}_{xx} + \mathbf{r}_x \times \mathbf{S}_{xx,i}\right)^{\mathrm{T}} \cdot \left(\mathbf{S}_{x,j} \times \mathbf{r}_{xx} + \mathbf{r}_x \times \mathbf{S}_{xx,j}\right) \\
\frac{\partial^2 g}{\partial e_i \partial e_j} &= 2 \cdot \mathbf{S}_{x,i}^{\mathrm{T}} \cdot \mathbf{S}_{x,j}
\end{aligned}
\tag{14}
$$

It should be pointed out that the items $\mathbf{S}_{x,i} \times \mathbf{S}_{xx,j}$, $\mathbf{S}_{x,j} \times \mathbf{S}_{xx,i}$ and $\mathbf{S}_{x,i}^{\mathrm{T}} \cdot \mathbf{S}_{x,j}$ only depend on the material coordinate of the integration point and the element dimension. They are not relevant to the current configuration, which means that they can be calculated at the pre-processing stage to save the calculation time.

## 4. Fractional Derivative Cable Damping

The traditional Kelvin–Voigt constitutive model is presented as follows [34]:

$$
\sigma(t) = E \cdot \varepsilon(t) + C \cdot \frac{\partial \varepsilon(t)}{\partial t}
\tag{15}
$$

In this model, the stress is divided into two parts associated with the strain and the changing rate of the strain, respectively. Bagley and Torvik introduced the three-parameter fractional derivative model, which can be stated as [35]:

$$
\sigma(t) = E \cdot \varepsilon(t) + C \cdot \frac{\mathrm{d}^{\alpha}\varepsilon(t)}{\mathrm{d}t^{\alpha}}
\tag{16}
$$

It can be seen that the changing rate of the strain in Equation (15) is replaced by the fractional derivative of the strain with respect to time. $\alpha$ is the fractional derivative order, whose value varies in the interval $(0,1)$. By picking $\alpha$ appropriately, the three-parameter fractional derivative model could perform better than the traditional viscosity model [30]. The fractional order derivative operator $\mathrm{d}^{\alpha}/\mathrm{d}t$ can be approximated by the Grünwald definition:

$$
\frac{\mathrm{d}^{\alpha}\varepsilon(t)}{\mathrm{d}t^{\alpha}} \approx (h)^{-\alpha}\sum_{j=0}^{N_t} A_{j+1}\varepsilon(t - jh)
\tag{17}
$$

where $N_t$ is the truncation number, $h$ is the time step of the integrator and $A_{j+1}$ is the Grünwald coefficient, given by a recursive form:

$$
A_{j+1} = \frac{j - \alpha - 1}{j}A_j, \;\; A_1 = 1
\tag{18}
$$

$\mathbf{C}$ is the viscosity coefficient matrix, which is obtained by multiplying an extra ratio $\tau$ by the elastic coefficient matrix $\mathbf{E}_v$. Therefore, the stress associated with the viscosity $\sigma_v$ could be expressed as:

$$
\sigma_v = \tau E_v \cdot \frac{\mathrm{d}^{\alpha}\varepsilon_{xx}(t)}{\mathrm{d}t^{\alpha}} = \tau E_v(h)^{-\alpha}\sum_{j=0}^{N_t} A_{j+1}\varepsilon_{xx}(t - jh)
\tag{19}
$$

Therefore, in this investigation, the energy related to the viscosity stress $U_v$ can be written as:

$$U_v = \frac{1}{2} \int_V \varepsilon_{xx}^{\mathrm{T}} \tau E_v(h)^{-\alpha} \sum_{j=0}^{N_t} A_{j+1} \varepsilon_{xx}(t - jh) \mathrm{d}V \tag{20}$$

Afterwards, the corresponding generalized viscous force $\mathbf{Q}_v$ can be obtained by taking the partial derivatives of the viscosity energy $U_v$ with respect to nodal coordinates $\mathbf{e}$:

$$
\begin{aligned}
\mathbf{Q}_v &= \frac{\partial U_v}{\partial \mathbf{e}} = \frac{\partial}{\partial \mathbf{e}} \left[ \frac{1}{2} \int_V \varepsilon_{xx}^{\mathrm{T}} \tau \mathbf{E}_v(h)^{-\alpha} \left( \sum_{j=1}^{N_t} A_{j+1} \varepsilon_{xx}(t-jh) + A_1 \varepsilon_{xx}(t) \right) \mathrm{d}V \right] \\
&= \frac{1}{2} \tau(h)^{-\alpha} \int_V \frac{\partial \varepsilon_{xx}^{\mathrm{T}}}{\partial \mathbf{e}} E_v \sum_{j=1}^{N_t} A_{j+1} \varepsilon_{xx}(t-jh) \mathrm{d}V + \tau(h)^{-\alpha} \int_V \frac{\partial \varepsilon_{xx}^{\mathrm{T}}}{\partial \mathbf{e}} E_v A_1 \varepsilon_{xx}(t) \mathrm{d}V \\
&= \mathbf{Q}_v(t-jh, t) + \mathbf{Q}_v(t)
\end{aligned} \tag{21}
$$

In order to use the implicit integrators to solve the system equation of motion, the second derivative of the viscosity energy with respect to nodal coordinates should be presented:

$$
\begin{aligned}
\frac{\partial \mathbf{Q}_v}{\partial \mathbf{e}} &= \frac{\partial \mathbf{Q}_v(t-jh, t)}{\partial \mathbf{e}} + \frac{\partial \mathbf{Q}_v(t)}{\partial \mathbf{e}} \\
&= \frac{1}{2} \tau(h)^{-\alpha} \int_V \frac{\partial^2 \varepsilon_{xx}^{\mathrm{T}}}{\partial \mathbf{e}^2} E_v \sum_{j=1}^{N_t} A_{j+1} \varepsilon_{xx}(t-jh) \mathrm{d}V \\
&+ \tau(h)^{-\alpha} \int_V \frac{\partial^2 \varepsilon_{xx}^{\mathrm{T}}}{\partial \mathbf{e}^2} E_v A_1 \varepsilon_{xx}(t) \mathrm{d}V + \tau(h)^{-\alpha} \int_V \frac{\partial \varepsilon_{xx}^{\mathrm{T}}}{\partial \mathbf{e}} E_v A_1 \frac{\partial \varepsilon_{xx}}{\partial \mathbf{e}}(t) \mathrm{d}V
\end{aligned} \tag{22}
$$

The second derivative of the strain with respect to nodal coordinates has been presented in Equation (12). It should be pointed out that the history of the strain is constant at the current time step. At the computer implementation stage, the viscous force can be calculated together with the elastic force because there are lots of duplicated terms in their expressions.

## 5. Computation Strategy

By introducing the generalized damping force proposed in the previous section into the traditional multibody system dynamic equations [36], one can obtain the system equation of motion:

$$
\begin{cases}
\mathbf{M}\ddot{\mathbf{e}} + \mathbf{Q}_e + \mathbf{Q}_v + \mathbf{C}_e^{\mathrm{T}} \boldsymbol{\lambda} = \mathbf{Q} \\
\mathbf{C} = \mathbf{0}
\end{cases} \tag{23}
$$

where $\mathbf{C}$ is the constraint equations, $\boldsymbol{\lambda}$ is Lagrange's multiplier and $\mathbf{Q}$ is the external force vector, including the gravity force and the contact force, etc. In multibody system dynamic simulation procedures, the constraint equations can be time-invariant, which is also called the continuity condition. Such constraints can be handled in the pre-processing stage. The Jacobian matrix of this type of constraint is constant so it can be used to identify independent and dependent variables, for example, the cantilever boundary condition used in Section 6.1 and the spherical joint used in Sections 6.2 and 6.3. Another type of constraint is time-varying which can only be solved along with the system equation of motion. The number of unknown variables of the system becomes large as Lagrange's multiplier is used [37].

The system equation of motion given in Equation (23) is a set of Index-3 differential-algebraic equations. In this investigation, the generalized alpha method is chosen to solve the system equation of motion as this method achieves an optimal combination of accuracy at the low-frequency range and numerical damping at the high-frequency range. Such a method exhibited good applicability to many tough problems [8]. More details about how to implement this integrator, including variable discretization and choosing appropriate solving parameters, can be found in the literature [38].

In this research, all the numerical examples are run in MATLAB R2021a. The program-running hardware is a Dell tower workstation. The processor is an Intel Xeon W-2235 with 3.8 GHz. The computer has 32 GB RAM, a 256 GB SSD and a 1 TB HDD. The operating system is Windows 10 Professional.

## 6. Numerical Simulation and Experimental Validation

### 6.1. Axial Stretch of Beam

A straight cantilever beam model composed of a 0.1% axial initial strain is modeled to demonstrate the application of the fractional derivative viscosity in the ANCF cable element. The initial configuration and the parameters of the model are shown in Figure 2. A 0.1% strain is initially applied on the beam so the free tip of the beam would experience longitudinal vibration. The x component of the position vector for the free tip is given in Figure 3, with the fractional order $\alpha$ set to be 0.5. As a comparison, the results obtained under different combinations of the parameter $\tau$ and the truncation number $N_t$ are shown. The results show that the vibration would be damped distinctly under the effect of viscous force, and the damping effect would become more intense with the increase in the ratio $\tau$ and the truncation number $N_t$. Figure 4 presents how the value of alpha affects the damping. Theoretically, when alpha is equal to 0, the pure elastic material is obtained. When alpha is equal to 1, the fractional derivative model will degenerate to the traditional Kelvin–Voigt constitutive material. From Figure 4, one can find that the damping effect becomes significant with the increase in the alpha value. It can be concluded that the fractional derivative viscosity model is apparently introduced into the ANCF cable element.

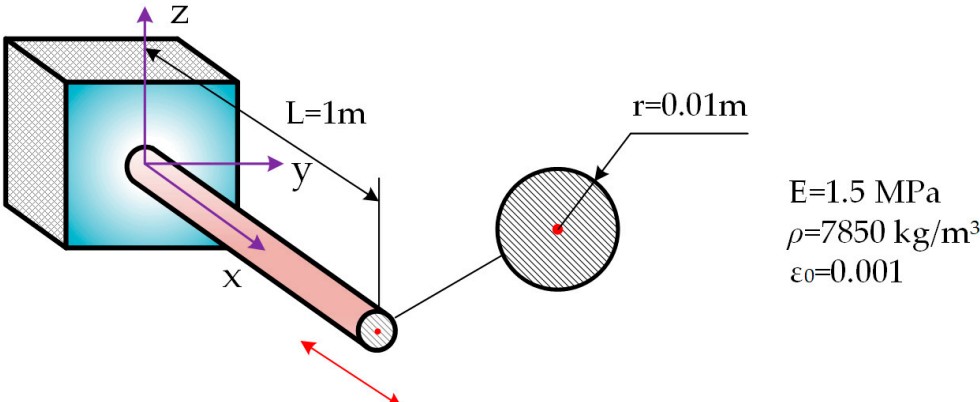

**Figure 2.** Cantilever beam model.

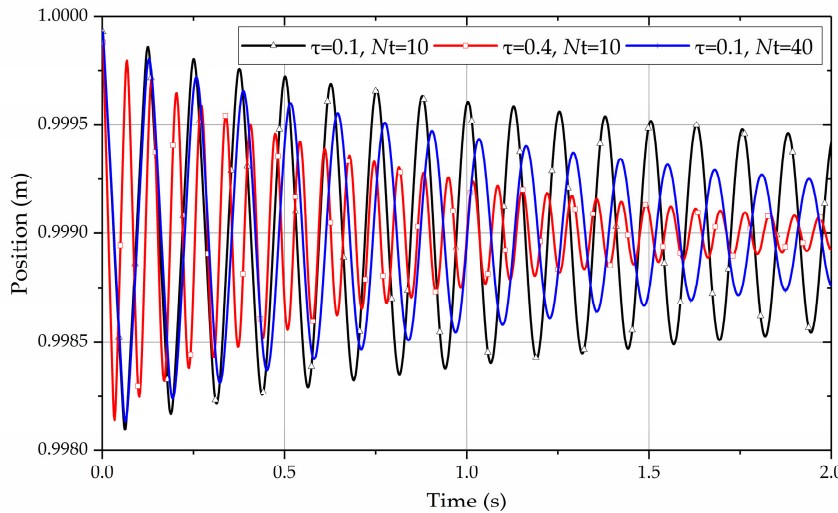

**Figure 3.** x-component of the position vector of the free tip with different parameters.

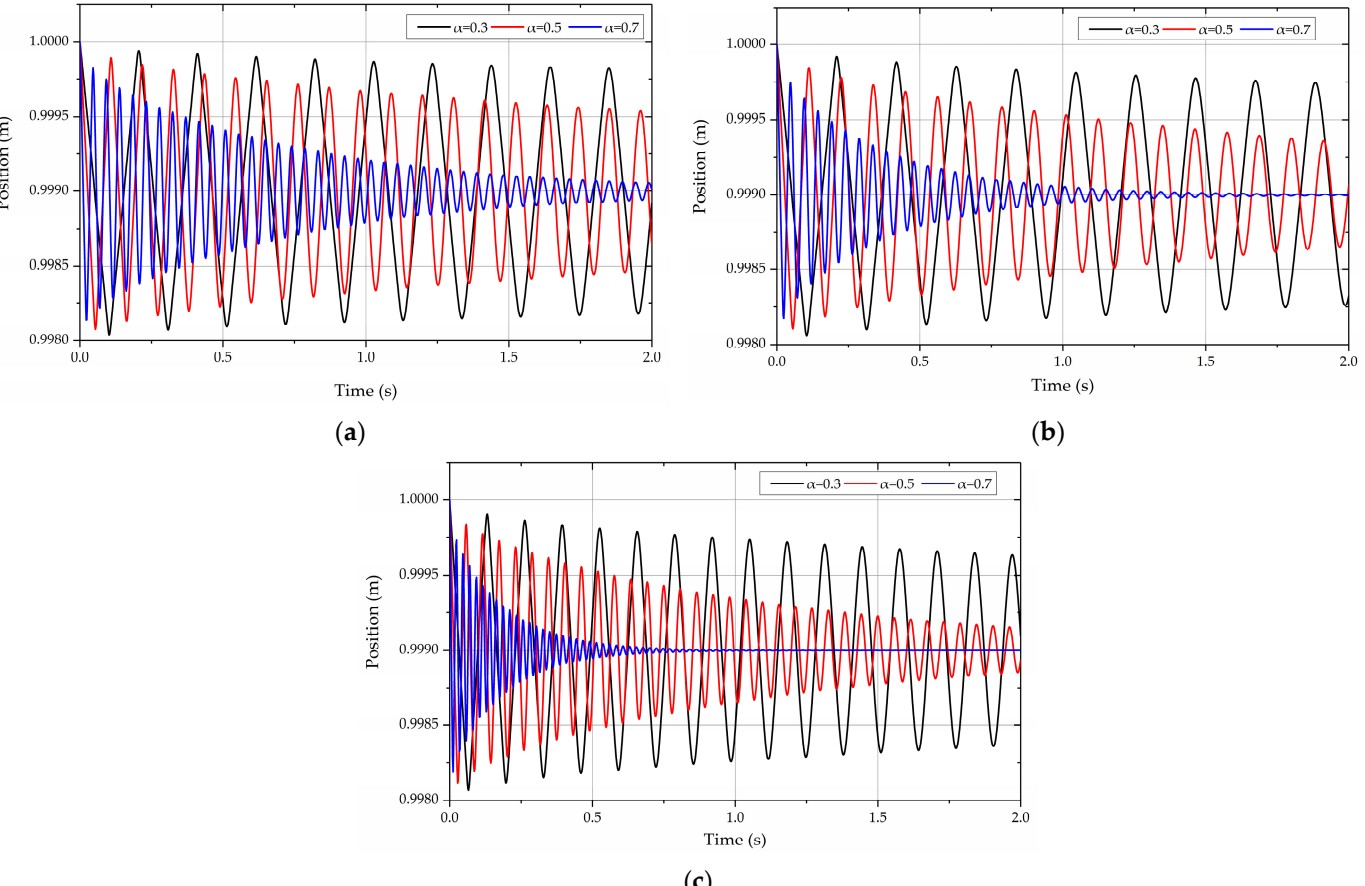

**Figure 4.** x-component of the free tip position at different derivative orders. (**a**) τ = 0.1, *N*t = 10; (**b**) τ = 0.1, *N*t = 40; (**c**) τ = 0.4, *N*t = 10.

### 6.2. Flexible Pendulum

In order to demonstrate the performance of the proposed cable damping model in the large overall motion and large deformation, a convergence test is performed on the flexible pendulum model shown in Figure 5. The material properties of the pendulum are given in Table 1. The simulation time is 1.5 s. The damping parameters used in this example are τ = 0.1, $N_t = 10$ and $\alpha = 0.5$. The pendulum is discretized into 4, 8 and 16 elements, respectively. The vertical displacements of the free tip and the total strain energy are compared in Figures 6 and 7, respectively. The configurations at different moments are shown in Figure 8.

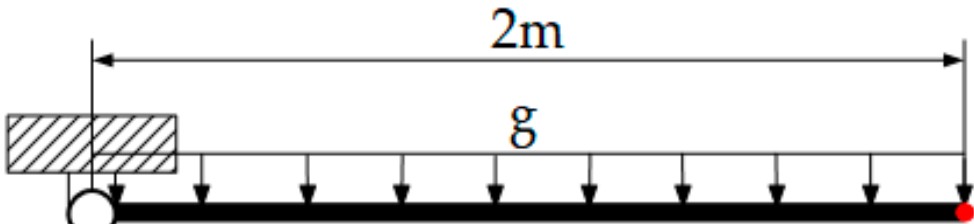

**Figure 5.** Flexible pendulum model.

**Table 1.** Parameters of the flexible pendulum.

| Properties | Length (m) | Radius (mm) | Gravity Acceleration (m/s²) | Density (kg/m³) | Young's Modulus (GPa) | Poisson Ratio |
|---|---|---|---|---|---|---|
| Value | 2 | 3 | 9.81 | 2320 | 63 | 0.33 |

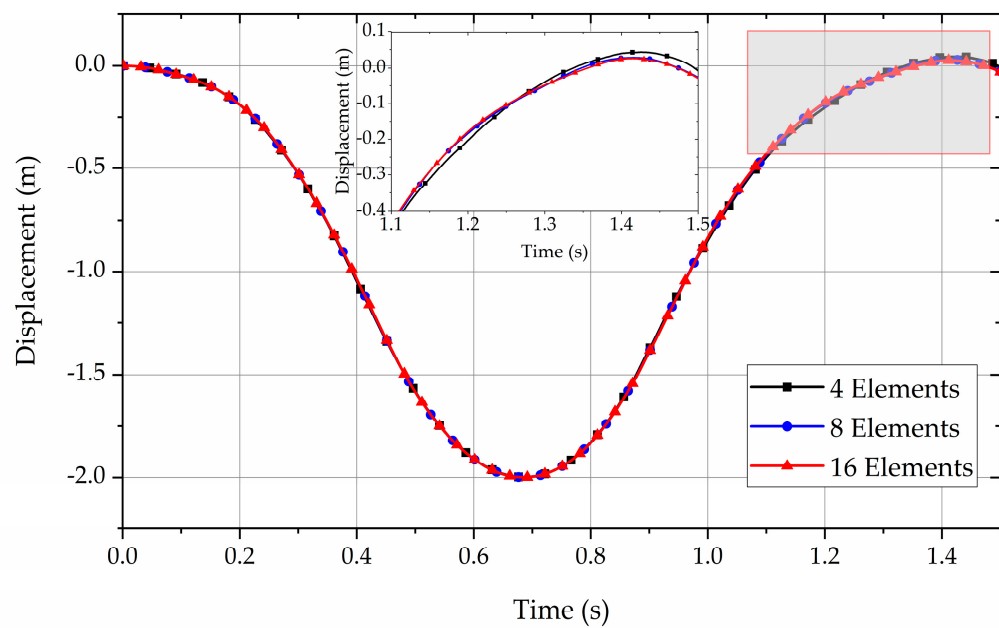

**Figure 6.** Vertical displacement of the free tip.

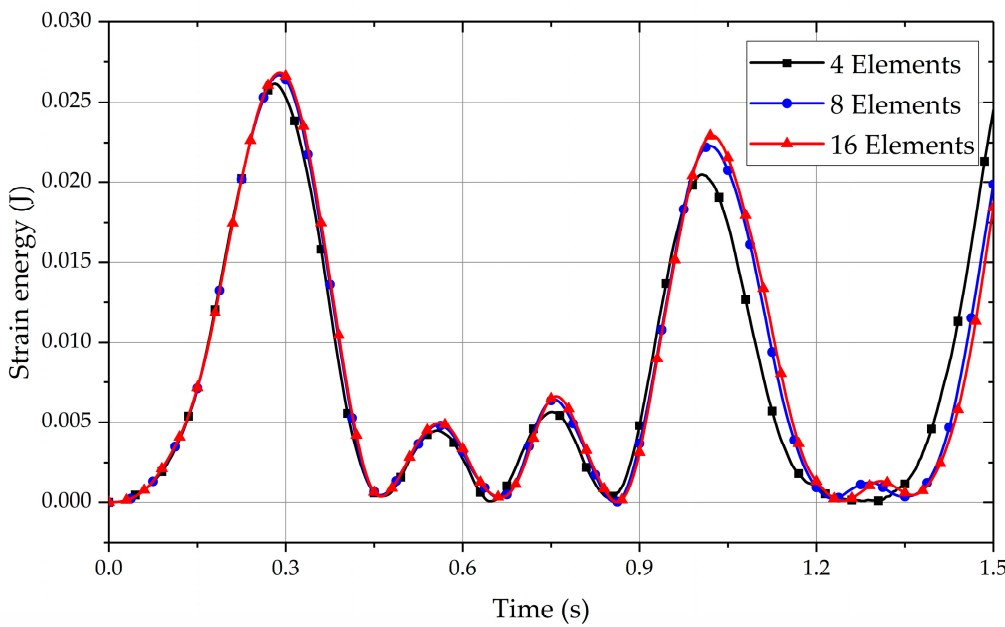

**Figure 7.** Total strain energy of the pendulum.

From these results, one can find that when increasing the number of elements, the curves become closer. It means that the convergence property of the cable element can still be preserved when the fractional derivative damping model is applied to the ANCF cable element.

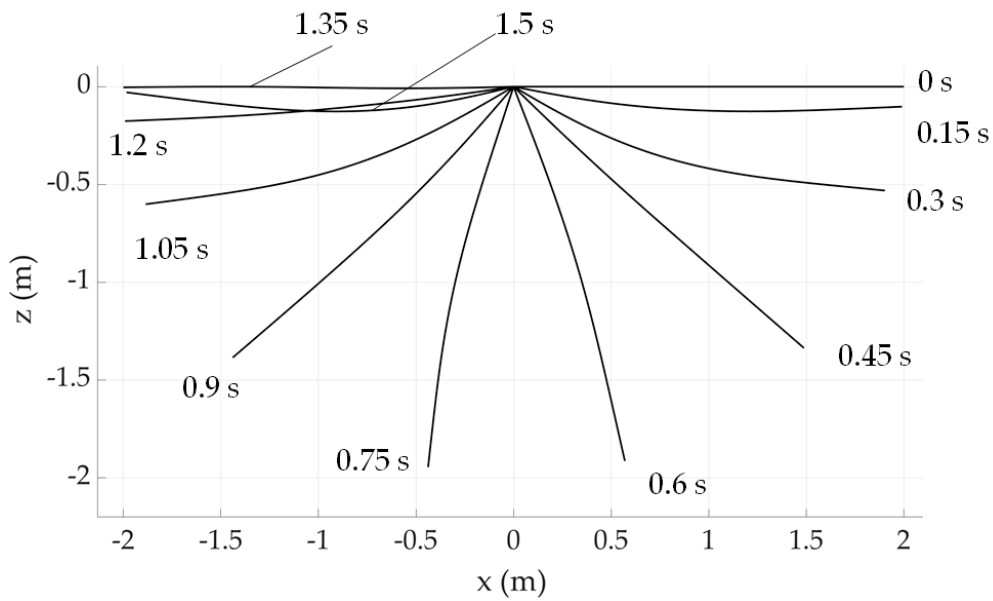

**Figure 8.** Configurations of the cable at different moments.

### 6.3. Wire–Sheave Contact Experiment and Computational Simulation

Wire rope failure is a kind of common issue in space engineering, which has a great influence on the control and safe operation of spacecraft. In this section, an experimental study is proposed to test the performance of the proposed fractional derivative cable damping model. Figure 9 gives the schematic view of the experiment. A wire is fixed at one end by a spherical joint, and the other end is tensioned and passes through two sheaves by a tensioner. A tension release device controlled by a wireless signal is connected in series in the wire, as shown in Figure 10. A tension sensor is also connected in series in the wire to record the tension force at the moment that the wire is released. When the wire is released, the timer starts, and the high-speed camera begins to capture the configurations of the wire. They will be compared with the numerical simulation results. It should be pointed out that the weight of the release device is large, making it difficult to keep the wire straight only by tension. Therefore, a load rope is set above the experimental wire to bear the weight of the wire releaser (as can be seen in the first subfigure of Figure 10).

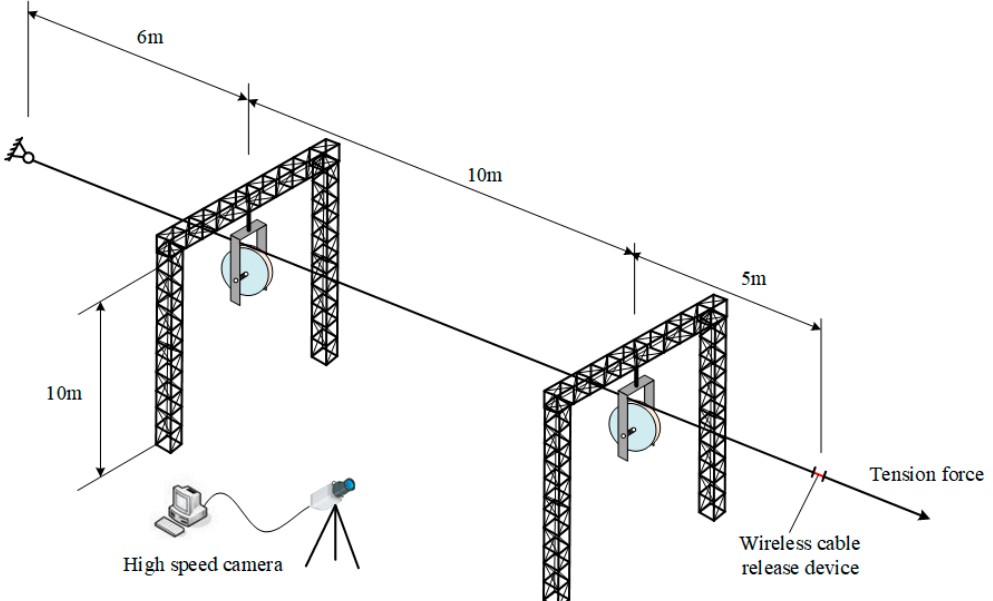

**Figure 9.** Schematic view of the experimental setting.

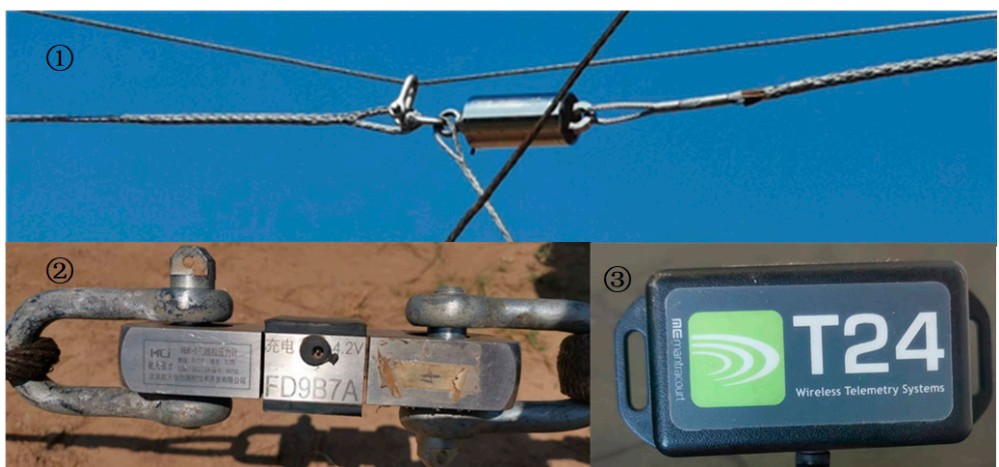

**Figure 10.** Some experiment devices (①release device,②tension sensor,③wireless communication module).

The wire was tensioned by a force of 7890 N, which is 10% of its breaking force. The experiment was conducted outdoors. The day of the experiment was clear and windless. The temperature was 28 degrees Celsius. At the computer implementation part, the static equilibrium configuration of the wire was solved and used as the initial configuration of the dynamic simulation [1,12]. The contact between the wire and the sheaves was implemented based on the penalty methods [8]. The simulation was performed for 4 s. Table 2 gives the geometry and material parameters of the wire. Figure 11 presents the comparative results of the wire configurations. They were obtained at 1 s, 2 s, 3 s and 4 s, respectively. The configurations from the simulation are colored by vertical velocity.

**Table 2.** Parameters of the wire.

| Properties | Length (m) | Radius (mm) | Density (kg/m$^3$) | Young's Modulus (GPa) | Poisson Ratio |
|---|---|---|---|---|---|
| Value | 21 | 13.45 | 2320 | 63 | 0.33 |

It can be observed that when the wire is released, it first retracts violently under the initial tension. Afterwards, under the influence of its own kinetic energy, it wraps around the right sheave at a large warp angle. Due to the large distance between the two sheaves, the wire falls, making it out of contact with the right pulley and continue to fall. One can find that the numerical simulation results reflect the dynamic process well. By comparing the key frame configuration between the numerical simulation results and the experimental results, it can be concluded that the simulation results reach good agreement with the experimental results. The feasibility of the proposed cable viscosity can be demonstrated. In spite of this, it should also be pointed out that in flexible multibody system dynamic experiments, comparing the configurations obtained from numerical simulations and experiments is still the dominant method. Any measurement of the contact of the flexible body itself will change its dynamic behavior. Therefore, in future work, we will try non-contact measurement methods that are more accurate than the configuration ratio, so that the experiment can better guide the theory.

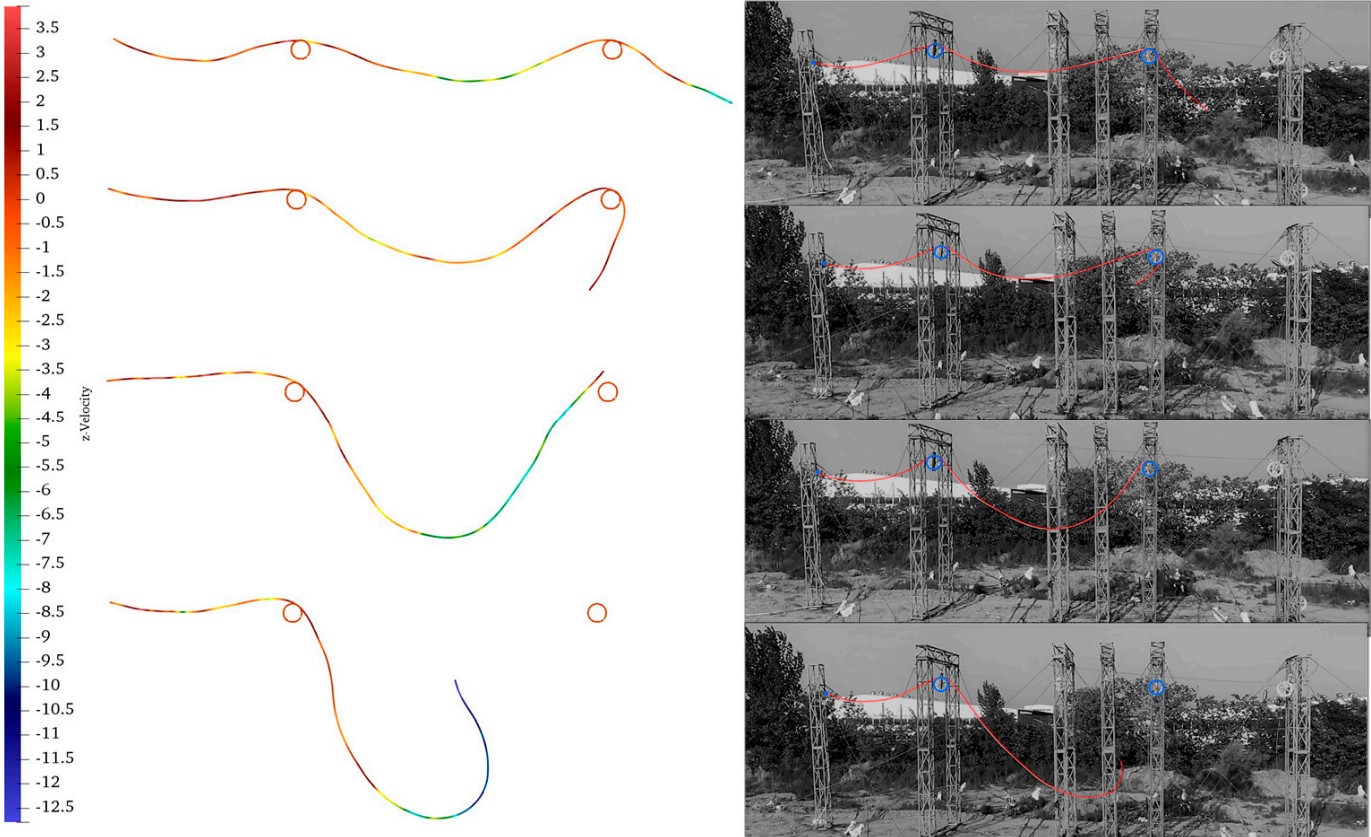

**Figure 11.** Comparison of the simulation and experimental results.

## 7. Conclusions

A novel viscosity model based on the fractional derivative material damping model is developed for the cables and wires discretized by the absolute nodal coordinate formulation. The original work of this investigation can be summarized as follows:

1. The fractional order derivative material damping model based on three-parameter formulation is introduced into the ANCF cable element. The generalized damping force and its Jacobian matrix with respect to the nodal coordinate are derived accordingly.

2. A cantilever beam with an initial longitudinal stretching strain is tested. The results show that the damping effect becomes more obvious with the increase in the viscoelastic coefficient $\tau$, truncation number $N_t$ and derivative order $\alpha$.

3. A soft pendulum model is checked to see the performance of the proposed damping model. It can be observed that when the number of elements used increases, the curves of the vertical displacement of the free tip and the total strain energy become closer. The convergence property is proved.

4. An experiment of wire tension release is performed. A wire goes through two sheaves and is tensioned by 10% of its breaking force. After it is released, it vibrates tempestuously and falls onto the sheaves. The configurations of the wire are captured by a high-speed camera and compared with the simulation results. The application of the proposed cable damping model based on the fractional derivative viscosity can be demonstrated.

The possible applications of the cable damping model in this paper include cable-driven mechanisms, spatial deployable structures, vibration characteristics of large steel structures with flexible cables, etc. In the authors' opinion, the work proposed in this investigation could be improved from the following two aspects. First, the viscosity was only developed for the stretch part of the cable element. It can be extended into the

curvature part. Second, a more precise experiment could be proposed to standardize the damping coefficients for the wire.

**Author Contributions:** Conceptualization, N.L. and Z.Y.; methodology, Y.G.; software, Y.G.; validation, P.L.; formal analysis, Y.G. and Z.Y.; investigation, Y.G.; resources, Y.G.; data curation, Y.G.; writing—original draft preparation, Y.G.; writing—review and editing, Y.G. and Z.Y.; visualization, Y.G.; supervision, N.L.; project administration, N.L.; funding acquisition, Z.Y. and P.L. All authors have read and agreed to the published version of the manuscript.

**Funding:** This research was funded by the National Natural Science Foundation of China, grant number 12272123; the Applied Fundamental Research Program of Changzhou, grant number CJ20220058; and the Independent Research Project of State Key Laboratory of Green Building in west China, grant number LSZZ202209.

**Data Availability Statement:** Not applicable.

**Conflicts of Interest:** The authors declare no conflict of interest.

## Nomenclature

| | |
|---|---|
| **r** | the global position of an arbitrary point on the cable |
| **S** | the shape function matrix |
| **e** | the nodal coordinate vector |
| **M** | the element mass matrix |
| U | strain energy of the ANCF cable element |
| E | Young's modulus |
| A | area of the cross-section |
| I | second moment of inertia of the cross-section |
| $\varepsilon_{xx}$ | the temperature and its gradient at the longitudinal direction |
| $K$ | curvature of the cable |
| $\mathbf{Q}_e$ | element generalized elastic force |
| $J_{i,j}$ | components in the Jacobian matrix of the elastic force |
| $\sigma$ | stress |
| $\varepsilon$ | strain |
| C | damping coefficient in Kelvin–Voigt constitutive model |
| $h$ | time step |
| $\alpha$ | fractional derivative order |
| $A_{j+1}$ | the Grünwald coefficient |
| $N_t$ | truncation number |
| $\tau$ | extra ratio |
| $\sigma_v$ | stress associated with the viscosity |
| $\mathbf{E}_v$ | elastic coefficient matrix |
| $\mathbf{Q}_v$ | generalized viscous force |
| Ce | Jacobian matrix of the constraint equation |
| **λ** | Lagrange's multiplier |

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
