# Peer review of "Fractional Derivative Viscosity of ANCF Cable Element"

_actuators, doi:10.3390/act12020064_

Round 1

Reviewer 2 Report

Dear Authors,

The manuscript attempts to implement a viscosity model for the cable damping problem. The problem statement is well defined and the method utilized is explained in detail.

However, some more details would improve the quality as below

1) Please mention what numerical discretization method did you use to convert the PDEs into your computational domain. Boundary conditions must be explicitly mentioned. 

2) Numerical error analysis is a must when a new equation is implemented in the computational space. Please add a section addressing this.

3) It seems like alpha was set to be 0.5 on section 5.1. What happens at other values and why was 0.5 chosen over other values? Please explain.

4) Need more references talking about other implementations to provide why your approach is novel.

Reviewer 3 Report

This is an interesting work, and the model and methods they developed are very solid. Hence, I suggest publishing this paper. However, some minor issues should be addressed before publishing.

1. The results appear to be marginal to the field of scientific research considered.

2. Please highlight how the work advances or increments the field from the present state of knowledge and provide a clear justification for your work.

3. The section Conclusions will be point out the original results of the paper and can be extended to highlight the contributions. Please provide a clear justification for your work in this section, and indicate uses and extensions if appropriate.

4.     Please check the paper again for any possible misprints.

5.     I think the author need to emphasize more clearly the contribution of the manuscript from a scientific point of view.

Reviewer 4 Report

Review of the paper:

Fractional derivative viscosity of ANCF cable element

Authors:

Yaqi Gu, Zuqing Yu, Peng Lan, Nianli Lu

The article presented for review is interesting from an application point of view. The absolute nodal coordinate formulation (ANCF) represents an interesting approach to modeling the dynamics of multibody systems with large deformations, which corresponds to the validity of the analysis made of the behavior of cables stretched between poles. Nonetheless, the paper submitted for review requires some clarifications and corrections in order to meet the requirements for a significant scientific publication in the Actuators Journal.

List of detailed comments below:

  1. The section Introduction: 1) lines 27-29: it appears that, due to the universality of the technical solution, the reader does not need to be given as many references for the practical applications of high-voltage cable; the indicated use in the section 5.3, lines 187-188, rather as a definition of the problem here should appear with a reference to the sources; 2) line 50: improve the description of the source - should be written [1].
  2. The legend descriptions for the variables used in the formulas are missing; the comment applies to most of the paper. Please correct.
  3. Section 4: presenting models: Kelvin-Voigt, Baglev and Torvik—please cite primary references.
  4. Section 5: Figure 2 – isn't it worth posting the coordinate system and giving the location of the point to be further analyzed?
  5. Section 5: Figure 3 – which coordinate refers to the position shown on the chart?
  6. Section 5: what computer program was used to numerically simulate the beam and pendulum?
  7. Figure 7: please complete the drawing with the height of the poles between which the rope is stretched; this is relevant to the experiment.
  8. Figure 8: no simulation testing environment is given. Where are these results from? Missing from the section is a broader discussion of the results obtained.
  9. The description of the experiment lacks an analysis of the stiffness of the poles between which the tested cable is stretched. The behavior of the cable may depend on the technical parameters of the system on which this cable is stretched.
  10. Section 6: The content of this chapter should be completely restructured. Much of the conclusions formulated could be the content of a separate subsection summarizing and discussing the results. The conclusion section should be more concise and specific.

Final conclusion: after reading, one gets the impression that the work was not done with due scientific care and that the content is somehow a piecing together of partial results from the work of earlier authors of the article. There is a lack of a "smooth transition" between the section presenting the analytical relationships used in the modeling and the simulation and experimental sections.

Round 2

Reviewer 1 Report

I carefully read the authors' responses to my comments. The authors put a lot of work into improving the article, but I still think they are presenting an imaginary problem "This numerical example is not designed for a real wire rope". In the opinion of the reviewer, scientific research is supposed to provide answers and guidelines for engineers (see scope "Acuators"). The numerical results cannot be compared to the experiment because I am afraid that the assumed data (see Tab. 2) is also unrealistic .

Reviewer 4 Report

Review of paper (second round):

Fractional derivative viscosity of ANCF cable element

Authors:

Yaqi Gu, Zuqing Yu, Peng Lan, Nianli Lu

In response to the criticisms indicated in the first review, the authors have mostly correctly addressed the comments indicated, correcting and supplementing the text of the paper. Most of the comments were discussed in the attached response file. The Introduction section, among others, was corrected and supplemented, citing a broader description of other researchers' contributions to the topic. In the "Materials and Methods" section, the legends of the variables used in the given mathematical relationships describing the physics of the phenomenon under study have been completed. However, there are still fragments that could be improved and supplemented:

1) Figure 2 – please record unit correctly - MPa.

2) The text further fails to specify the environment used for the simulation study.

3) For a better comparison of the differences in the results obtained, you can make graphs of the results presented in one chart (note applies to Figure 4-6).

4) Figure 11 shows a view of the mounting sensors on the rope – Doesn't such sensor mounting change the behaviors of the cable during the experiment? After all, in these places is not a continuous system, the stiffness, and damping of the system change. Please discuss the indicated problem, its impact on the results, and the possibility of solving (bypassing) this problem.

5) The result in Figure 12 is not comprehensively described and discussed in the text of the paper.

Round 3

Reviewer 1 Report

Generally, mathematicians examining the stability of a given method (e.g. integration) take arbitrary, sometimes unreal parameters. And in this case it is justified.

In mechanical engineering, when we talk about "a wire rope", we expect real parameters (not the density equal to the density of water -1000kg/m^3 and the Young's model 1 MPa, the value corresponding to ??).

The measure of the correctness of a method is rather a comparison of the method with the experiment.

Assuming the linear behavior of the wire rope in the dynamics analysis is an ill-posed task in advance.

  Adoption of small values does not guarantee that the method will be correct for their larger values.

In conclusion, I appreciate the contribution made to improve the paper, especially the "controversial" example. I believe that the paper can be judged by the Actuators readers.